# Using Systems Dynamics for Capturing the Multicausality of Factors Affecting Health System Capacity in Latin America while Responding to the COVID-19 Pandemic

**DOI:** 10.3390/ijerph181910002

**Published:** 2021-09-23

**Authors:** Kathya Lorena Cordova-Pozo, Hubert P. L. M. Korzilius, Etiënne A. J. A. Rouwette, Gabriela Píriz, Rolando Herrera-Gutierrez, Graciela Cordova-Pozo, Miguel Orozco

**Affiliations:** 1Institute for Management Research, Radboud University, P.O. Box 9108, 6500 HK Nijmegen, The Netherlands; hubert.korzilius@ru.nl (H.P.L.M.K.); etienne.rouwette@ru.nl (E.A.J.A.R.); 2Dirección Técnica de Servicio Médico Integral, Luis Alberto de Herrera 2275, Montevideo 11600, Uruguay; gabriela.pirizet@gmail.com; 3Centro Integral de Medicina Familiar CIMFA, Caja Nacional de Salud, Villa Galindo, Calle La Merced s/n., Cochabamba, Bolivia; dr.rolandohg@gmail.com; 4Unit of anaesthesiology, Hospital Seton Caja Petrolera de Salud, Km 5 Av. Blanco Galindo, Cochabamba, Bolivia; gracielacordova@gmail.com; 5Department of Pathophysiology, Facultad de Medicina, Universidad Mayor de San Simón, Av. Aniceto Arce s/n, Cochabamba, Bolivia; 6Freelance Consultant in Health and Development, Managua, Nicaragua; maorozcov@gmail.com

**Keywords:** group model building, system dynamics, COVID-19, Latin America, multicausality, exploratory factor analysis, mediating effects

## Abstract

Similar interventions to stop the spread of COVID-19 led to different outcomes in Latin American countries. This study aimed to capture the multicausality of factors affecting HS-capacity that could help plan a more effective response, considering health as well as social aspects. A facilitated GMB was constructed by experts and validated with a survey from a wider population. Statistical analyses estimated the impact of the main factors to the HS-capacity and revealed the differences in its mechanisms. The results show a similar four-factor structure in all countries that includes public administration, preparedness, information, and collective self-efficacy. The factors are correlated and have mediating effects with HS-capacity; this is the base for differences among countries. HS-capacity has a strong relation with public administration in Bolivia, while in Nicaragua and Uruguay it is related through preparedness. Nicaragua lacks information as a mediation effect with HS-capacity whereas Bolivia and Uruguay have, respectively, small and large mediation effects with it. These outcomes increase the understanding of the pandemic based on country-specific context and can aid policymaking in low-and middle-income countries by including these factors in future pandemic response models.

## 1. Introduction

Since 2019, the severe acute respiratory syndrome coronavirus 2 (SARS-CoV-2, mostly known as COVID-19) caused over 192 million infections and over 4.2 million deaths worldwide [1]. While the World Health Organization has a framework for the management of an emergency outbreak, which focused mainly on the health system [1,2], other countries chose to emphasize in building trust between the population and the government [3]. However, most pandemic response plans relied on the monitoring of infections and deaths using some form of the SIRD model (susceptible, infectious, recovered, and deaths) to predict trends over time in order to avoid a collapse of the healthcare system [4,5,6]. The SIRD model led to useful and standardized pandemic response, mainly in the form of restrictions, that enhanced global security [4], assuming a direct relation between the enactment of the measure and its acceptance by the population.

Governments took the lead in responding to the pandemic by introducing COVID-19 restrictions (e.g., staying at home, taking 1.5 m distance and allowing only limited group sizes for special events such as weddings and funerals; from now on these are referred to as restrictions) to prevent virus transmission and a collapse of the healthcare system. Over time, different outcomes were visible across countries [7]. Richer countries managed to control the pandemic longer with restrictions supported by financial compensations for their population and companies, while poorer countries took longer to offer smaller compensations to the population since they depended on international aid (e.g., International Monetary Fund and the World Bank emergency funding) [8]. It is possible the delayed response is the reason why Latin America is among the worst affected by the pandemic [9]. As well, the informal economy played a role in the pandemic outbreak [9,10]. The informal economy is the part of the economy that is not monitored by the government, and therefore it is nonexistent [11]. On this continent, informal employment ranges between 30% and 80% and informal companies have low savings, capital and investment rates and were often excluded from any type of assistance during COVID-19 due to the informality [12]. For many individuals, it was impossible to close their business and stay at home, as their subsistence depended on daily work. Finally, a large part of the population in Latin America has no free access to health care [9,10,13]. This contributed to a higher COVID-19 transmission [12,14] which was not projected by SIRD models.

Latin American countries have many similarities (e.g., language and culture) but also show large differences (e.g., education, and health system) both between and within countries (e.g., urban versus rural, main cities versus small ones) [13]. This may have influenced the results of the COVID-19 pandemic response. Whilst Uruguay and Nicaragua seemed to have the pandemic under control with a low number of cases, Bolivia saw an increasing number of cases and a health system collapse despite strict restrictions [15]. While most Latin countries prolonged the restrictions to better control the increasing number of infections, other countries had almost no restrictions (e.g., Nicaragua and Brazil). Yet, both options led to adverse impacts to the economy with high infections and deaths and a lengthy quarantine. Hence, the pandemic progression was inconsistent with the generic SIRD results.

Therefore, to improve the pandemic monitoring and response, the general SIRD model needs to be complemented with additional variables that adequately capture the meaningful differences between countries. For this purpose, system dynamics (SD) is well suited as it helps to identify the network of variables that drives behavior over time [16]. A reference mode of behavior is a pattern related to historical evidence, which can be time series data or anecdotes [17,18]. SD can extend beyond linear causal modeling by visualizing the complex interrelationships between contextual factors and public health interventions like the COVID-19 pandemic management where *mediators* and *moderators* operate together as mechanisms to affect the health system capacity (from now on referred to as HS-capacity) [19]. Clarifying the feedback structure and mechanisms responsible for changes in infection rates over time helps to understand the restriction effects and improve policymaking [17,18]. The objective of this study was to capture the multicausality of factors affecting HS-capacity while responding to the COVID-19 pandemic. We aimed to identify meaningful similarities and differences between countries in response to COVID-19. We focused on three Latin American countries (Bolivia, Nicaragua, and Uruguay) to examine the key mechanisms in terms of causal relationships and feedback loops that next to SIRD play a central role in planning more effective response. This approach considered health and social aspects and their interactions, which is in line with the World Health Organization (WHO) perspective on health which comprises both physical, mental, and social well-being.

## 2. Materials and Methods

Three countries were chosen because they represent an archetypical range of countries in Latin America in terms of income and informal economy [12,20]. For example, one could say Nicaragua and Bolivia are like El Salvador or Paraguay, and equivalents of Uruguay are Chili or Costa Rica. Over time, each country experienced different outcomes concerning COVID-19, in numbers of infections and deaths, during the period of March until December 2020 when this study was conducted. Three methods were used: group model building (GMB), exploratory factor analysis (EFA), and correlational and mediation analyses. Together, these served to validate the GMB-results for each country.

### 2.1. System Dynamics (SD)–Group Model Building (GMB)

The aim of SD is to capture the current situation and how it evolved from past conditions, visualizing the causal structure that explains the observed behavior by eliciting and integrating the knowledge of experts involved in the COVID-19 pandemic. This in the core consists of a set of interrelated variables and factors that together form a feedback-loop structure [17,18]. Feedback loops can be of two types: reinforcing loops (R) that accelerate initial behavior, and balancing loops (B) that move toward an equilibrium. GMB is a facilitated process of SD that graphically maps the implicit and explicit knowledge of participants (i.e., decision makers, experts [21] and other stakeholders) and portrays the full breadth of the problem while other methods only study specific parts. GMB is useful when a theoretical structure or conceptual model of the complete system is not defined [22], as was the case with COVID-19. We used GMB to construct qualitative models in the form of causal loop diagrams (CLDs) which visualize the system’s variables and their relations. By identifying causal relations between variables, options for change are identified that could be the base for policymaking [22,23,24,25,26]. Some benefits of this qualitative approach are that it allows to elicit and combine the ideas of participants by visualizing the underlying structure of the problem. The modelling process helps to integrate different perspectives and contextual details that interactively shape the model. A downside of using qualitative models is that it is impossible to simulate. In this study, three GMB projects in three Latin American countries were conducted to create insight into the COVID-19 dynamics.

A multidisciplinary team designed the GMB sessions and a subsequent survey. It was composed of an expert GMB modeler, a mathematical modeler, a health economist and four medical doctors. The team helped checking the consistency of the model, the approach to potential stakeholders, and the surveys in every country. Stakeholder selection was based on purposeful sampling combined with a snowball technique to recruit health personnel who experienced the impact of COVID-19 in their own region, as well as other inclusion criteria (Appendix B). A diverse group participated in the GMB sessions [27], ensuring a large variation of perspectives in the data collection [26,28]. In total, 20 stakeholders participated: seven in Bolivia, seven in Nicaragua, and six in Uruguay. Each received an invitation by email explaining the goal of the study, the questions, and how the results were going to be used. Invitees were informed about the ethical aspects and confidentiality of information to which they agreed and signed an informed consent.

For each country, two GMB online sessions were organized using Zoom, Miro, and Vensim software to communicate and sketch the dynamic model as recommended by Wilkerson et al. [29]. The process included individual tasks (e.g., eliciting ideas), group discussion, and iterative modelling to achieve consensus, such as capturing ideas in terms of causes and consequences, identifying factors that could be controlled (control variables), and performance indicators (target variables) [21]. The experts in the health systems discussed why the restrictions were not reducing the number of infections as expected, represented graphically by the reproduction rate [30] and how the pandemic was impacting the HS-capacity. The questions used to evoke the model structure were: How has COVID-19 impacted the healthcare system? What factors explain the COVID-19 infections over time and what can be done to reduce infections? What would improve the situation? The model was designed and elaborated during the sessions in direct interaction with the participants and was later checked for consistency. The sessions were summarized in a document for the feedback and acceptance of the stakeholders.

### 2.2. Validation Phase

The next stage was the appraisal of the GMB to test their fit to other data on the local situation, which is a basis for using the models to support policymaking in response to COVID-19. Three steps were used to assess validity and reliability. *First*, the main factors (*public administration* for the management of COVID-19, *preparedness*, *information,* and *collective self-efficacy)* and their relationships with HS-capacity. These factors were developed initially in an inductive manner in the modelling sessions, and then validated through multiple methods and data sources: internal consensus (within each country), patterns across countries, and were checked against scientific literature to support external validity [31]. *Second*, a country-specific online-survey was designed to check the validity of relations addressing the factors: with HS-capacity by grouping questions by factor like *The elements that help the management of COVID-19 are?* This was followed by a list of elements such as comprehensive epidemiological data, planning, and coordination. The contribution of each element was measured with a 5-point Likert scale ranging from ‘strongly disagree’ = 1, to ‘strongly agree’ = 5. The survey was disseminated in October 2020 via an online social-network within the health personnel in each country with a purposive snowball sampling. The factors can be found in later in the results and Appendix A, Appendix B, Appendix C and Appendix D. A total of 276 questionnaires were completed (20 invitees denied participation).

*Third*, an exploratory factor analysis (EFA) based on the online-survey was used to test if the relationships among these factors and the HS-capacity were supported by empirical data. EFA is a causal modeling method used to quantitatively explore the relation among variables and underlying factors. For EFA we used a bootstrapping resampling method (5000 replications) to produce robust approximations for small samples [32,33] and because GMB models may have nonlinear relationships [34]. Once we obtained the underlying factors and validated them, correlational and mediation analyses were applied by means of SPSS (Version 27).

## 3. Results

This section presents a simplified GMB with a brief explanation of the main factors, and EFA presents the validation per country.

### 3.1. GMB Models

All countries show the same balancing loop mechanism between HS-capacity and the epidemiological stock where an increase of infected people directly reduces the HS-capacity, and this results in an increase of the number of deaths by COVID-19. Countries differ with regard to the other four factors (public administration, preparedness, information and collective self-efficacy) which we describe below. All countries had a different reference mode of behavior over time represented by the reproduction number of infections (Rt). Every country implemented restrictions to reduce the spread of infections which was growing (Rt > 1, in blue) and had to decrease (Rt < 1, in red) (Figure 1, Figure 2 and Figure 3).

#### 3.1.1. Bolivia

Figure 1 shows the following. The upper part of the model shows the management of COVID-19 by public administration is part of a reinforcing feedback loop represented by R1. A high inflow of COVID-19 patients decreases HS-capacity creating a supply-demand discrepancy, which could be lowered by good pandemic management. Public administration comprises planning and coordination of three elements: (1) the collaboration among the government, private and scientific community; (2) bringing to bear previous epidemic experiences; and (3) the informal economy that negatively affects the number of infections and deaths. The management is indirectly influenced by “political quality” (e.g., degree of centralization, collaboration) that motivates or demotivates the collaboration and may delay achieving pandemic control. Reduced collaboration negatively influences the strategy of preparedness leading to a delayed increase in the HS-capacity. Centralized decisions often discouraged (private public) collaboration and this was reinforced in the pandemic.

Preparedness (R2). A prompt response to COVID-19 requires preparedness. Its resulting behavior depends on the amount and the quality of information received, which is an important driver for HS-capacity. Preparedness includes having timely protocols, rapid epidemiological information, and rapid tracing of COVID-19, but in the case of Bolivia each of these elements suffered from delays because of a centralized management of COVID-19.

Information (balancing feedback loop, represented by B1). When the need for information, both by the population as well as the health care personnel, outgrows what is available, official information is needed to fill the gap. However, as official information was limited (e.g., tests, epidemiological information by region), misinformation overflowed the system, creating a balancing loop. Sources included people posting “their own solutions to COVID-19”, influencing preparedness and collective self-efficacy.

Collective self-efficacy (R3). If demand for health care increasingly exceeds supply, this contributes to a higher level of collective self-efficacy. There are many paths to distinguish in this loop. One of them is the reinforcing loop of people taking their own measures to avoid death of COVID-19 starting with the (1) higher number of deaths that maintains awareness of the problem, (2) the informal economy that reduces collective self-efficacy, (3) reduced access to the health care system especially for inhabitants of small towns or rural areas, and for the health personnel the higher risk at work because of lack of biosecurity measures. As information was missing, the actions were reflected in the social media to share their “own recipes” for COVID-19, and returned back as misinformation which affected the HS-capacity.

Participants identified three main leverage points: the reduction of centralized management of COVID-19 pandemic, increased access to information (as well as education to prepare and read the information), and the increase access to healthcare.

#### 3.1.2. Nicaragua

Figure 2 displays the following. Public Administration (R1). In Nicaragua, the main governmental strategy in response to COVID-19 is denial of the problem, which necessitates the creation of a reinforcing loop of centralization. Centralization was present before the pandemic and was reinforced during the pandemic. Its implementation increased three aspects: (1) fear-based working environments; (2) personnel that is laid off if they mention that the situation is not under control; and (3) non-strengthening of the first level of care. Therefore, there were no data available on COVID-19, nor was there a collaborative environment that would increase HS-capacity.

Preparedness (B1 and R1). Participants named two factors as important for an efficient response to COVID-19: (1) the strengthening of first level of care, and (2) adopting protocols. B2 indicates a negative reinforcing loop that shows the lack of access to free biosecurity materials or the necessary inputs. In addition, participants indicated there was neither an environment of respect nor collaboration, which negatively affects HS-capacity. R1 indicates that an increased discrepancy between supply and demand for HS-capacity increased the flow to (black market) pharmacies, or traditional medicine to avoid dying from COVID-19. Participants mentioned that primary healthcare was not prepared to respond to COVID-19 and HS-capacity did not increase.

Information (B3). The implementation of “reduced official information” increases misinformation, including stories on the actions of the population to avoid contagion and death from COVID-19. Official information directly reinforces the epidemiological stock and indirectly informs protocols that improve the response and the HS-capacity.

Collective self-efficacy (R2). The higher the discrepancy of supply-demand in the HS-capacity, the higher the collective self-efficacy. Access to health care was always limited which meant inhabitants had to rely on themselves, a situation which was reinforced by the pandemic Two important factors that contributed to increased self-efficacy were: (1) the number of deaths that kept awareness high, (2) the limited capacity to self-quarantine due to poverty and informal economy, obliging people to take their own measures to avoid COVID-19 using self-medication (natural medicine or from pharmacies). For the health personnel, the major factor was the fear-based working environment or prohibition to use biosecurity material. Together, this resulted in a high number of people turning to pharmacies or black markets of medicaments to prevent COVID-19. This was reflected in the use of social media posting information on actions that needed to be taken to avoid COVID-19 which came back as misinformation. This affected the HS-capacity through preparedness.

Finally, participants identified one main leverage point, which was the reduction of centralized management of COVID-19 pandemic.

#### 3.1.3. Uruguay

Figure 3 shows the following. Public Administration (R2). An increasing mismatch between supply and demand for healthcare increases the pressure on government to come up with a clear strategy to respond to COVID-19. The management of the pandemic relied on an increase in the co-responsibility and collaboration between different public and private organizations. The reinforcing loop created trust of the population seeing that all was managed well. This trust allowed the society to follow the imposed restrictions which increased the HS-capacity.

Preparedness (R4). Participants indicated the strategy of “*decongestion in the hospitals and readiness for the emergency*”. This included a balancing loop driven by an intense use of information, high collaboration, co-responsibility, and trust as the existing elements in the culture of the country that were reinforced by the pandemic. In combination helped to increase HS-capacity. The first level of care is accessible from all over the country, included a follow-up of patients at home or brigades using different sources of information like phones, telemedicine, and other applications. Information played a key role in monitoring the situation and ensured HS-capacity was at its highest level.

Information (R1). Again, if demand for information outstrips supply, the need for information and use of technology will increase. Experts thought that the effectiveness of governance pushed the collaboration of private, public, and scientific sectors to create and buy software, use of apps, and telemedicine that enabled to respond COVID-19 pandemic in time.

Collective self-efficacy (R3). Uruguay was ready for a pandemic but at the time of the GMB sessions no crisis had yet occurred. Therefore, the reduced discrepancy between supply and demand in the health system, reduced the need for collective self-efficacy. At the same time, preparedness reinforced the trust of the population which made them follow government guidance, increasing HS-capacity.

The participants identified three leverage points to intervene in the system: (1) border control, as inhabitants from neighboring countries tried to get in to receive free treatment, which potentially could increase the pandemic; (2) reducing work pressure and demotivation of the health personnel that was working hard to avoid an outbreak; and (3) counteracting the loss of credibility of the population for the “exaggerated restrictions” that could result in people trying to get back to “normal” without following the government’s guidance.

#### 3.1.4. Main Outcomes

The models for each country showed a similar structure with factors driving HS-capacity while differentiating in the context variables. The literature supports the main pathways visualized in the country-models, presenting elements with conceptual direct relations with HS-capacity (Table 1).

### 3.2. EFA: Comparative Analysis and Validation

As presented, each GMB system displays four clusters of variables that drive and are driven by HS-capacity during the COVID-19 pandemic. These factors and their interrelationships were tested by means of surveys among health personnel. We used an exploratory factor analysis (EFA) with Oblimin rotation and principal axis factoring. The EFA confirmed a four-factor structure (eigenvalues > 1, factor loadings ranged between 0.39 and 0.93) and internal consistency of each scale with a high-level level of reliability (Table 2). See Appendix C for the factors, items, means and standard deviations.

We explored the factor structure identified by EFA to test if the structure of the GMB could be replicated. Common method bias is a potential problem here as our data are gathered in a self-completed survey. A Harman’s single-factor test was used on items selected by EFA with no rotation and results were compared against the threshold of 0.50 [50]. This resulted in one factor explaining 30.76%, 30.38%, 45.80% of the total variance for Bolivia, Nicaragua, and Uruguay, respectively. Consequently, we can tentatively conclude that common method bias was not a major issue. Then, we used a Kaiser-Meier-Olkin (KMO) test and Bartlett sphericity test which indicated a good reliability and validity (Table 2) and the structure in each country confirmed that individual factors resulting from the GMB could be identified. Then, Cronbach’s alphas were calculated for every construct and for the survey indicating a good degree of consistency (values ranging from 0.77 to 0.96). In general, Cronbach’s alpha values should be 0.70 or higher although values of 0.60 or higher are also considered acceptable [51]. Finally, we calculated the construct reliability (CR) and the average variance extracted (AVE) to assess the composites of the factors. Fornell and Larcker (1981) suggest a threshold of 0.60 for CR, and 0.50 for AVE [52]. For both, similar or higher values were achieved with few exceptions, demonstrating the constructs measure the intended concept (CR), and have little variation in the items within the construct (AVE). 

#### 3.2.1. The Correlations among Factors

A correlation analysis was used to identify interrelations between the factors. Figure 4 presents Pearson correlations for the study factors (public administration, preparedness, information, collective self-efficacy) that positively correlate with HS-capacity. The presence of collective self-efficacy was mainly due to HS-capacity collapse, which occurred in Bolivia and Nicaragua (Figure 4).

#### 3.2.2. Mediating Effects

The GMB models show that preparedness is an important mechanism driving changes in HS-capacity. However, each model structure also shows that preparedness in turn is dependent on public administration that determines the strategy in response to the pandemics, such as the level and duration of restrictions. Public administration facilitates the availability of information that eventually affects collective self-efficacy, which in turn affects HS-capacity (thus, closes the loop). Accordingly, to assess mediating effects, we tested the impact of both preparedness and public administration on HS-capacity through the mediation of information and collective self-efficacy, respectively. We also tested all possible mediation effects and only report the ones that are statistically significant and common to all three countries. All three countries had positive significant mediation effects through information in Bolivia and Uruguay and through collective self-efficacy in Nicaragua (Figure 5).

The statistical analyses confirm three similar characteristics with the GMB models. First, this validates the factor-structure displayed by the GMB. Second, the factors are correlated with HS-capacity. Third, there are mediating effects on HS-capacity by these factors, particularly the information and the collective self-efficacy. The differences in the GMB are partially replicated through the analyses of correlation and mediation. Both analyses show that in Bolivia, public administration has a strong relation with HS-capacity, while in Nicaragua and Uruguay are related through preparedness. Nicaragua lacks information as a mediation effect whereas Bolivia and Uruguay have, respectively, small and large mediating effects with HS-capacity.

## 4. Discussion

People around the world expect their governments to come up with a response to the COVID-19 pandemic that is both effective and in line with local needs, considering health as well as social aspects. Previous studies used a generic SIRD model to respond to the pandemic, while to understand developments on a country-level, we need to include the country-specific context. Our study reveals the multicausality of factors responsible for fluctuations in HS-capacity while responding to the pandemic in Bolivia, Nicaragua, and Uruguay. These factors are similar across the three countries shaping a similar structure affecting HS-capacity while differentiating in the context variables.

An increasing number of COVID-19 infections reduces HS-capacity (epidemiological stock) which causes deaths [5,30]. The level of preparedness for the pandemic directly affects HS-capacity and depends on public administration for the management of COVID-19. This is done through official information or misinformation. Next, HS-capacity is impacted through the collective self-efficacy of the population who feel to have the abilities to stay healthy when they perceive the HS-capacity is collapsing. This is confirmed by the literature that found conceptual direct linear relations between HS-capacity and public administration [9,14,35,40,41,42,43,44], preparedness [6,7,35,45], information [38,39,46,47,48], or collective self-efficacy [14,41,49,50].

An EFA model was used to assess the structure designed in the GMB and validated the interrelations between factors and variables in all three countries. Mediation analysis assessed the mechanisms that are country-specific. This showed that the path coefficients were significant from the preparedness to HS-capacity in Nicaragua and Uruguay, and from public administration to HS-capacity in Bolivia. Information or collective self-efficacy are good mediators of preparedness or public administration to increase its effect on HS-capacity. Nicaragua lacks information as a mediator, whereas in Bolivia and Uruguay information has, respectively, small and large mediating effects with HS-capacity. This context is in line with real-world observations where the society’s behavior and the misinformation dominated the COVID-19 response in countries where the perceived HS-capacity was collapsed [13,41].

Political aspects are indeed an important part of the explanation. In this study, we differentiate between countries by mentioning “centralized” and “decentralized” management of the COVID-19 pandemic, Bolivia and Nicaragua being “centralized” while Uruguay is “decentralized”. Centralization negatively affected the outcomes in terms of lower preparedness, insufficient information, and a larger need for collective self-efficacy. Centralization did not stimulate collaboration or learning loops that could benefit HS-capacity, maintaining a fear-based working environment in Nicaragua and ignoring previous outbreak-experiences in Bolivia. This is in line with Liang et al. who established a negative relation between governance effectiveness and mortality in 169 countries [47] and Mazzucato and Kattel [38] who conceptually revealed that collaboration and trust helped in improving the COVID-19 response. Uruguay showed more feedback loops amongst factors with a focus on preparedness driven by co-responsibility, trust and collaboration between sectors, institutions, health personnel and the population, plus an intense use of technology. This is roughly in line with a study carried out in South Korea confirming the need for clear information, robust public administration, and citizenship cooperation [48]. The experience of Uruguay supports the need to work on aspects of inter-institutional collaborations, promote trust and co-responsibility as key elements to increase the information flow and a comprehensive pandemic response based on multi-disciplinary actor base [38,46]. This management reduced the need to rely on collective self-efficacy that negatively impacted HS-capacity in Bolivia and Nicaragua.

Methodological value. SD offers the possibility to visualize the whole situation, the factors, and interconnections. However, there is little evidence to identify quantitatively the useful elements found in the GMB [17]. Therefore, this study constructed facilitated GMB models by local health experts and validated them with a survey with the general population. This implies the possibility of developing a conceptual model using GMB that facilitates a broad understanding of the problem, describing the underlying mechanisms and resulting dynamics of the effects of earlier states of the health system to subsequent states [16]. Using the EFA method as a validation tool opens the possibility to validate complex nonlinear models diagrammed in GMB sessions. To our knowledge, the EFA as part of the structural-equation-modeling (SEM) was used for COVID-19 [50] but not yet as a validating method for GMB models. The choice of using both methods could help policy makers to move beyond healthcare and away of linear causality [16] to better address epidemics and creating comprehensive response-approaches while including the stakeholders’ perspectives [20,21,22,25].

Limitations and assumptions. The study has four main limitations. First, only people that were knowledgeable about the healthcare system participated. This may have caused bias in two ways. One, with similar opinions (however, by selecting individuals from different institutions and backgrounds, we think the bias has been mitigated). In addition, by using a follow-up questionnaire in larger populations of the health personnel, we partially corroborated the mechanisms revealed by the stakeholders which indicates that bias did not overly influence our results. Two, this excluded the vision of others like economists and social-welfare specialists, who could have shed light on the impact on systems closely related to health care. Future research could take them into consideration to cover their perspectives on these important issues. Second, the time for collecting the survey was limited to a month in times of the pandemic, which led to fewer completed questionnaires than expected. Third, the study used EFA and mediating analysis but other methods like SEM could validate more directly the GMB results, but this would have required large data samples [51]. Fourth, generalization of our results to other Latin American countries should be conducted with caution as this was exploratory research which may not have revealed generic patterns applicable for the whole continent. However, these limitations have to some extent been covered by the multiple validations of the study’s findings which partly supported each other and as such made the conclusions of this study robust. 

Our contribution is threefold. First, it adds empirical evidence for three Latin American countries. Second, it tests all similar factors of the structure with its mechanisms as a system while differentiates the context variables. Third, it adds a SD perspective to health-literature.

## 5. Conclusions

Our results show that the SD-perspective enabled the stakeholders in three Latin American countries to translate their expertise in GMB, exhibiting the multicausality of factors affecting HS-capacity during the COVID-19 pandemic. It showed dynamic patterns explaining health outcomes which were partly validated by means of extant literature and through questionnaires among health personnel. The use of SD captured the similarities and differences in the three countries while responding to the COVID-19 pandemic. In our models, similar mechanisms pertaining to information and collective self-efficacy were observed in response to COVID-19. Differences were witnessed in relation to the role of preparedness and public administration. Overall, including the revealed factors may foster more effective policymaking and future pandemic response in Latin American countries.

## Figures and Tables

**Figure 1 ijerph-18-10002-f001:**
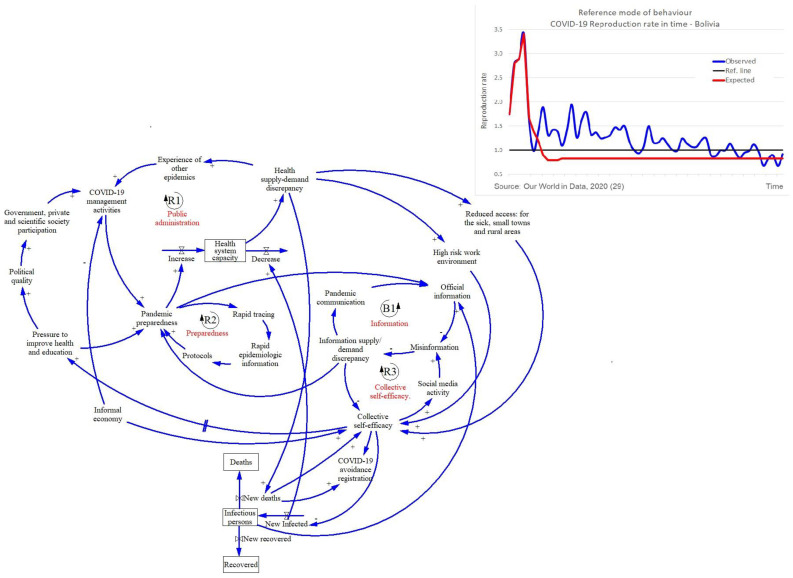
Reference mode and GMB Bolivia. Solid lines indicate positive relationships while a dotted lines indicate negative relationships. Reinforcing feedback loops are represented by an R, and balancing feedback loops are represented by a B.

**Figure 2 ijerph-18-10002-f002:**
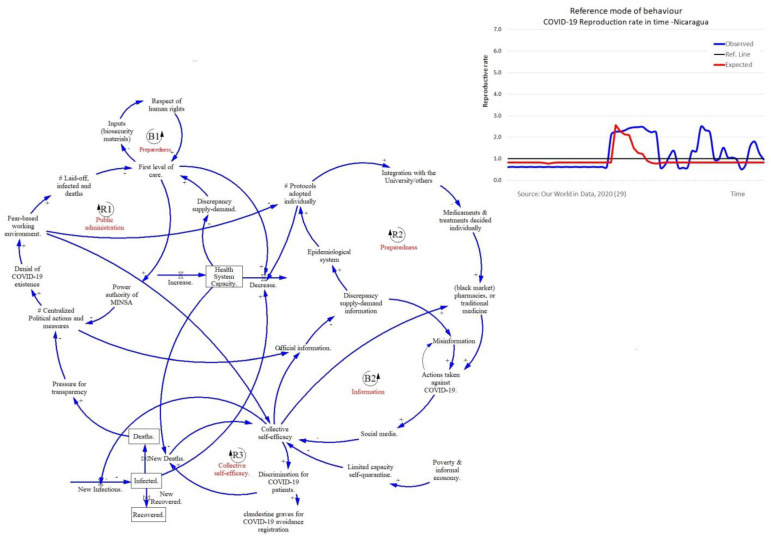
Reference mode and GMB Nicaragua. Solid lines indicate positive relationships while a dotted lines indicate negative relationships. Reinforcing feedback loops are represented by an R, and balancing feedback loops are represented by a B.

**Figure 3 ijerph-18-10002-f003:**
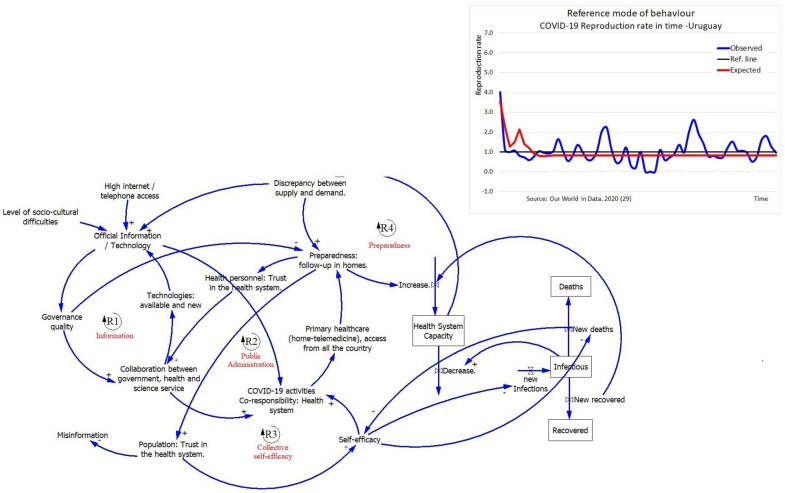
Reference mode and GMB Uruguay. Solid lines indicate positive relationships while a dotted lines indicate negative relationships. Reinforcing feedback loops are represented by an R, and balancing feedback loops are represented by a B.

**Figure 4 ijerph-18-10002-f004:**
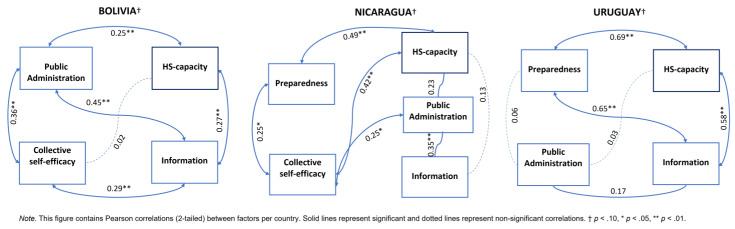
Factor correlations for each country. This figure contains Pearson correlations (2-tailed) between factors per country. Solid lines represent significant and dotted lines represent non-significant correlations. † *p* < 0, * *p* < 0.05, ** *p* < 0.01.

**Figure 5 ijerph-18-10002-f005:**
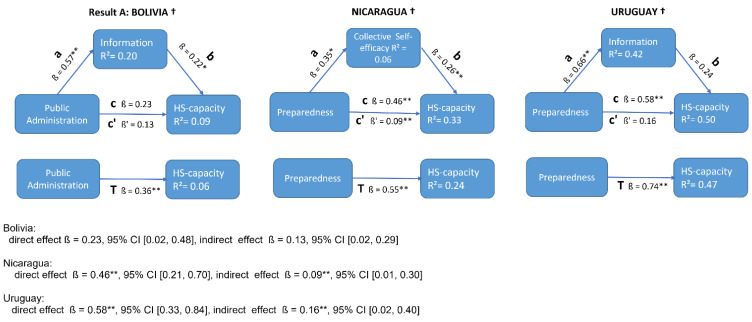
The mediating effect of information and collective self-efficacy. Details: ab = mediation effect; c = direct effect; c’ = indirect effect; T = total effect. The number of bootstrap samples was 5000. ß = direct effect and ß’ = indirect effect. † * *p* < 0.05, ** *p* < 0.01.

**Table 1 ijerph-18-10002-t001:** Main factors to categorize codes relevant to the health system capacity.

Systemic Feedback Mechanisms	Main Factors	Description of the Factor
Health system capacity	Public administration	Refers to resources and the capacity to collaborate between the State, the civil society, and the market (scientific community, universities, private and public health system) to stop the spread of COVID-19 while addressing the economic needs of the country [14,35]
	Preparedness	According to the World Health Organization (2019), preparedness refers to a framework to manage multisectoral disaster risk management, and all-hazards emergency preparation and response, including for epidemics, health systems strengthening and community-centered primary health care [36]. In our view, it refers to a set of actions which are aimed to decrease pandemic outcomes, such as protocols, biosecurity measures, rapid tracing, use of epidemiologic information and communication.
	Collective efficacy	Self-efficacy means the responsibility of an individual to behave careful or be effective in a pandemic. By collective efficacy we refer to a group’s shared perception that together they can stay healthy during the COVID-19 [37], it supports actions that could help avoid being infected by COVID-19.
	Information & Misinformation	The official information refers to messages by the government or formal sources to support the management of the pandemic (timely and sufficiently detailed). Misinformation fills the lack of detailed information or knowledge that according to the Mills, et al. (2020) this is done through: (1) distrust of science or selective use of expert authority; (2) distrust in pharmaceutical companies and government; (3) straightforward explanations; (4) use of emotion; and, (5) information bubbles [38]

**Table 2 ijerph-18-10002-t002:** Exploratory factor analysis and reliability per scale.

Description	Health System Capacity	Public Administration	Preparedness	Information	Collective Self-Efficacy
Literature supporting the direct relation to health system capacity	[4,5,9,39]	[9,14,35,40,41,42,43,44]	[6,7,35,45]	[38,39,46,47,48]	[14,41,49,50]
BOLIVIA					
Cronbach’s alpha for the whole survey	0.82				
Kaiser-Meyer-Olkin	0.76, *p* < 001				
Cronbach’s alpha for every construct	0.87	0.87		0.67	0.74
Factor Loadings	0.87–0.89	0.46–0.84	not applicable	0.43–0.70	0.39–0.81
NICARAGUA					
Cronbach’s alpha for the whole survey	0.84				
Kaiser-Meyer-Olkin	0.68, *p* < 001				
Cronbach’s alpha for every construct	0.78	0.96	0.68	0.78	0.89
Factor Loadings	0.86–0.93	0.49–0.85	0.43–0.75	0.56–0.82	0.84–0.90
URUGUAY					
Cronbach’s alpha for the whole survey	0.95				
Kaiser-Meyer-Olkin	0.76, *p* < 001				
Cronbach’s alpha for every construct	0.82	0.69	0.95	0.92	
Factor Loadings	0.45–0.69	0.52–0.77	0.62–0.90	0.53–0.86	not applicable

## Data Availability

Data supporting reported results can be obtained by contacting the corresponding author.

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
