# Peer review of "Using Systems Dynamics for Capturing the Multicausality of Factors Affecting Health System Capacity in Latin America while Responding to the COVID-19 Pandemic"

_ijerph, 2021, doi:10.3390/ijerph181910002_

Round 1
Reviewer 1 Report
This is a fascinating attempt to understand the response to Covid at the national level.
Sample: 135 How were the stakeholders identified? Please identify their positions in their healthcare systems. Epidemiologists, infectious disease physicians.... You do not provide your sampling frame. However, the three countries chosen should have published lists of such experts % coverage that your achieved.
168 you used "purposive snowball sampling." The snowball is likely to include more individuals with similar opinions and information than a random sample. How many individuals were identified per source? E.g., did source A identify 6 additional people for the sample and Source B 5.?...This provides information on the degree to which the sample depends on a limited number of people.
Results
You have studied these macro factors to understand how major societal decisions about resources and structure (and the finances to support these changes) need to be changed. Governments and experts from your data would not be able to estimate the range of reliability of your model as analysed to know how certain they could be that changes would result in improvements and be able to justify changes and expenditures. Can you provide a range of reliability estimates for funders (governments, World Bank...) and other change agents?
In Table 1 is "% of variance" the % of variance explained by your model? If so why are the % variances so low and can you say what factors determine the remaining % of variance?
For the items with the lower Cronbach-alpha scores, for Bolivia Information is 0.67 (0.43-0.70) and collective efficacy 0.74(0.39-0.81), for Nicaragua preparedness 0.68 (0.43-0.75), for Uruguay public administration 0.69 (.52 -0.77). Considering the wide range of intervals and the lower limits of each of these items, please provide an estimate of the overall range of reliability of your entire model
One typo 38 infected; change to infections
Author Response
Response to Reviewer 1 Comments
This is a fascinating attempt to understand the response to Covid at the national level.
Thank you very much for your inspiring words. We have addressed your comments to the best of our knowledge.
Point 1: Sample: 135 How were the stakeholders identified? Please identify their positions in their healthcare systems. Epidemiologists, infectious disease physicians.... You do not provide your sampling frame. However, the three countries chosen should have published lists of such experts % coverage that your achieved.
Response 1: People were involved in each of the three countries with an extensive network in the health sector. They suggested potential stakeholder who were approached by means of purposeful sampling combined with a snowball technique to recruit health personnel who experienced the impact of COVID-19 in their own region, as well as other inclusion criteria that are detailed in Appendix 2. The positions are included in Appendix 2 as presented below.
Appendix 2. Experts’ characteristics in the GMBs
Point 2: 168 you used "purposive snowball sampling." The snowball is likely to include more individuals with similar opinions and information than a random sample. How many individuals were identified per source? E.g., did source A identify 6 additional people for the sample and Source B 5.?...This provides information on the degree to which the sample depends on a limited number of people.
Response 2. For the GMB sessions we indeed used purposive snowball sampling as mentioned in Point 1. Each co-author in each country identified about an equal number of participants for the GMB session, resulting in a total of 7 participants for Bolivia, 7 for Nicaragua and 6 for Uruguay. You are right that this may have biased selection, resulting in a set of individuals with more similar opinions and information than in a random sample. However, by selecting individuals from different institutions and backgrounds, we think the bias has been mitigated. In addition, by using a follow-up questionnaire in larger populations of the health personnel, we partially corroborated the mechanisms revealed by the stakeholders which indicates that bias did not overly influence our results. We have included this point in the discussion, line 459-464.
Point 3: Results
You have studied these macro factors to understand how major societal decisions about resources and structure (and the finances to support these changes) need to be changed. Governments and experts from your data would not be able to estimate the range of reliability of your model as analysed to know how certain they could be that changes would result in improvements and be able to justify changes and expenditures. Can you provide a range of reliability estimates for funders (governments, World Bank...) and other change agents?
Response 3: This is a very interesting comment that we could use as a starting point for future study. This current study does not focus on how this analysis could results in improvements in the health system at the level of funding. We think that for this, more detailed additional modeling would be necessary. This modeling would be based on the macro factors discussed in the present study.
Point 4: In Table 2 is "% of variance" the % of variance explained by your model? If so why are the % variances so low and can you say what factors determine the remaining % of variance?
Response 4: The percentages of explained variance in Table 2 are related to the test of common method bias, which is a potential problem in survey research. For this purpose, we tested by means of Exploratory Factor Analysis (EFA) whether this was a problem. Basically, this tests how much variance is explained by a single factor, representing the common method bias, in all variables per country while employing no factor rotation method (this procedure is known as the Harman Single-Factor Test; Podsakoff, MacKenzie, & Podsakoff, 2003). In our study, the results were below the threshold for of 0.50 (Field, 2005) suggesting that common method bias would not be a problem in our data. To avoid confusion, we have removed this information from the Table 2 and just discussed it in lines 341-345.
Point 5: For the items with the lower Cronbach-alpha scores, for Bolivia Information is 0.67 (0.43-0.70) and collective efficacy 0.74(0.39-0.81), for Nicaragua preparedness 0.68 (0.43-0.75), for Uruguay public administration 0.69 (0.52 -0.77). Considering the wide range of intervals and the lower limits of each of these items, please provide an estimate of the overall range of reliability of your entire model
Response 5: In general, Cronbach’s alpha values should be .70 or higher although values of 0.60 or higher are also considered acceptable (Griethuijsen et al., 2014). Especially, for newly developed scales this latter threshold is used and most of our reported values are in accordance with this, although in future research this requires additional attention. The figures that you are referring to in your Point 5 are factor loadings which basically represent the correlation of items with the underlying factors. Factor loadings of 0.30 and higher are usually considered sufficient (Field, 2005) and all our factor loadings meet that criteria. In response to your question, we have added the range of all factor loadings in the text. We hope this answers your concern.
Point 6: One typo 38 infected; change to infections
Response 6: Thank you, we modified this in the text.
References
Podsakoff, P. M., MacKenzie, S. B., & Podsakoff, N. P. (2003). Common method biases in behavioral research: A critical review of the literature and recommended remedies. Journal of Applied Psychology, 88(5), 879-903.
Field A. Andy Field - Discovering Statistics Using SPSS. Vol. 58, Journal of Advanced Nursing. 2005. 303–303.
Griethuijsen, R. A. L. F., Eijck, M. W., Haste, H., Brok, P. J., Skinner, N. C., Mansour, N., et al. (2014). Global patterns in students’ views of science and interest in science. Research in Science Education, 45(4), 581–603.

Reviewer 2 Report
Thank you for the opportunity to review your manuscript about strategic interventions to mitigate the Covid 19 pandemic.
Background: You mention informal economy. What does this term mean?
Since the pandemic is a long lasting, over time, and we have not seen the end of this pandemic. Therefore, you need to define under which period of time this study was conducted. Also, it would be helpful to mirror the covid -19 situation (covid cases, spreading rate, death) in each country selected for this study during this time. Maybe, this could be presented in a figure or table.
Line 97 and forward (Three methods were used….) should be moved to the methods section.
Methods: Your method section is well described. However, since I´m not familiar with the methods you have used, I have no further comments on this section.
Results: The figures are well adjusted to the text presentation, and the results seems accurate.
Discussion: Your discussion is fine. However, I miss the political aspects- are there major differences between political structures and startegeic decitons made between the countries that could also explain the pandemic process? I also think about the term preparedness, within the HS organizations. What does it actually mean? Economical preparedness; equipment; organizational structures; competence?
Conclusion: Consider to “lift” the conclusion section a bit, by moving the results presented to the discussion section and focus on overall conclusions.
Round 2
Reviewer 1 Report
Thank you for your edited manuscript. It was really easy to see the changes you made. The key to your article is the reliability and validity of your data sources, and you provided comments agreeing with my concerns. Please include your replies as changes in your manuscript and not just to the reviewer - this will strengthen it.
Author Response
Dear Reviewer 1. Thank you for your time in reviewing the document. We have addressed all your comments and included them in the text as announced in our precedent reply to you. I am sorry to not have sent the document to you before for your verification.
Please see the attached document with track changes, addressing those aspects that you had concerns with.
The English language and style were also verified with a native speaker.

Reviewer 2 Report
Nothing further to comment
Author Response
Thank you very much for your review and for the approval of the document to be published.
The English language and style of the document has been reviewed by a native speaker.